# PORF-DDPG: Learning Personalized Autonomous Driving Behavior with Progressively Optimized Reward Function

**DOI:** 10.3390/s20195626

**Published:** 2020-10-01

**Authors:** Jie Chen, Tao Wu, Meiping Shi, Wei Jiang

**Affiliations:** The College of Intelligence Science and Technology, National University of Defense Technology, Changsha 410073, China; chenjie@nudt.edu.cn (J.C.); wutao@nudt.edu.cn (T.W.); weijiang@nudt.edu.cn (W.J.)

**Keywords:** autonomous driving, progressive optimization, deep reinforcement learning, reward function, sequential frames

## Abstract

Autonomous driving with artificial intelligence technology has been viewed as promising for autonomous vehicles hitting the road in the near future. In recent years, considerable progress has been made with Deep Reinforcement Learnings (DRLs) for realizing end-to-end autonomous driving. Still, driving safely and comfortably in real dynamic scenarios with DRL is nontrivial due to the reward functions being typically pre-defined with expertise. This paper proposes a human-in-the-loop DRL algorithm for learning personalized autonomous driving behavior in a progressive learning way. Specifically, a progressively optimized reward function (PORF) learning model is built and integrated into the Deep Deterministic Policy Gradient (DDPG) framework, which is called PORF-DDPG in this paper. PORF consists of two parts: the first part of the PORF is a pre-defined typical reward function on the system state, the second part is modeled as a Deep Neural Network (DNN) for representing driving adjusting intention by the human observer, which is the main contribution of this paper. The DNN-based reward model is progressively learned using the front-view images as the input and via active human supervision and intervention. The proposed approach is potentially useful for driving in dynamic constrained scenarios when dangerous collision events might occur frequently with classic DRLs. The experimental results show that the proposed autonomous driving behavior learning method exhibits online learning capability and environmental adaptability.

## 1. Introduction

With the development of artificial intelligence technology [1,2,3,4,5], autonomous driving has made considerable progress. End-to-end autonomous driving vehicles, which use machine learning algorithms to generate control policies directly from sensor perception information are different from traditional autonomous driving vehicles with environment perception [6,7], path planning [8], and vehicle control [9]. The end-to-end autonomous driving technology is similar to the human driving mode and has attracted more and more attention for decades. As early as 1988, an unmanned vehicle named ALVINN [10] used a multilayer perceptron to learn the direction control of the vehicle. In 2005, Yann LeCun et al. [11] used a 6-layer Convolutional Neural Network (CNN) with RGB images of the binocular camera as the network input and the steering angle as the output to learn vehicle obstacle avoidance in off-road. In 2016, in the work of Bojarski et al. [12], a DNN was used to map images to 3 degrees of freedom (DOF) control commands (steering, throttle, and braking) to control the vehicle, which proved that DNN could learn the control process of driving from raw image data. In 2017, Codevilla et al. [13] proposed the condition imitation learning model whose inputs included not only perception information but also high-level navigational instructions so that the learning model could drive according to the instructions autonomously. In 2019, Amini et al. [14] introduced a variational end-to-end navigation and localization model that used raw camera data of the environment as well as higher-level roadmaps to learn how humans drive in residential areas.

DRL is a representative of AI technology. AlphaGo [15] and Alpha Zero [16], the Go programs based on DRL, have been able to completely defeat the human professional players of the highest level. DRL provides a solution for the perception and policymaking of unmanned vehicles, making it possible for models to learn autonomous driving behavior end-to-end. Lange et al. [17] used the top view images of the track for toy vehicle control model training. A pre-trained autoencoder was used to extract the features of the images, and then a cluster-RL-based Deep Q Learning method was used for control model training. The training achieved good results and could exceed the control level of human players. In 2016, Sallab AE et al. [18] proposed to apply the end-to-end trained DRL model to lane-keeping assistance. By training and comparing the Q learning algorithm and the Deep Deterministic Actor–Critic (DDAC) algorithm on the virtual game engine TORCS, it was proved that the DDAC algorithm with continuous action output could achieve better control effects and smoother vehicle trajectories. In 2017, Hyunmin Chae et al. [19] applied DQN to the vehicle autonomous braking control system. The trained DQN control model was tested in a virtual environment, and the results showed that the model performed well in many braking conditions with uncertainties. In 2018, Kyushik et al. [20] proposed to apply DRL to highway driving conditions. The trained model was tested in a virtual environment, and the results showed that the vehicle could drive efficiently in a simulated highway scene and avoid collisions with other traffic participants. In 2019, Wayve [21] in Cambridge pre-trained the DRL model in a virtual environment to achieve lane tracking and then fine-tuned the network in the real environment. This work was based on sparse rewards, which will inevitably lead to low training efficiency and failure to converge to the optimal state.

According to the above works, although DRL has shown great potential in many aspects, the application of DRL in autonomous driving is not outstanding compared with other fields. The reason is that the training of DRL is a process of trial-and-error. In the training stage, it needs to collect various behaviors that unmanned vehicle may make in various states, and determine the corresponding rewards or punishments according to the consequences of these behaviors. In real scenarios, unmanned vehicles will inevitably face safety risks such as collisions when collecting various behaviors. Therefore, DRL can improve the behavior of unmanned vehicles effectively in a virtual environment, but its performance in a real environment becomes unsatisfactory. To solve these problems, an autonomous driving behavior learning method based on a progressively optimized reward function is proposed. This method can provide dense and reasonable immediate rewards for the DRL model and enables the model to be trained with a continuously updated training sample set by means of human-machine collaboration in the real environment. Thus, the autonomous driving behavior can be gradually optimized to enhance the environmental adaptability of the vehicle.

### Contribution and Novelty

First, a progressive optimized reward function (PORF) learning model using a Deep Neural Network (DNN) is proposed for representing driving adjusting intention by the human observer in the loop. The DNN-based reward model is progressively learned using the front-view sequential dynamic images as the input and via active human supervision and intervention.

Secondly, PORF is integrated into the Deep Deterministic Policy Gradient (DDPG) framework, i.e., PORF-DDPG, for learning personalized autonomous driving behavior in a progressive learning way. In PORF-DDPG, the experience replay memory pool representing human driving experience can be updated automatically via human-machine collaboration for achieving continuous learning of the DDPG model. Consequently, PORF-DDPG exhibits the ability to progressively learn personalized driving behaviors in various scenarios.

Finally, experimental validation from the virtual environment to the real environment is performed that the unmanned vehicle equipped with PORF-DDPG can achieve the progressive learning ability of personalized autonomous driving behavior and adaptability in different road scenarios.

## 2. The Progressive Learning Method of Autonomous Driving Behavior

In this section, we first outline the structure of PORF-DDPG and introduce the relationship between each part. Then, we analyze the design ideas and structure of each component separately and introduce the model training process.

### 2.1. Overview of PORF-DDPG

In this paper, an autonomous driving behavior learning model based on the DRL algorithm is built as the end-to-end control model for unmanned vehicles, which can generate control policies directly from the vehicle states. These policies include direction and speed instructions, both of which can be regarded as selecting actions from a limited continuous action space. At the same time, the vehicle states include velocity, steering, camera information, etc. thus, the state space is almost unlimited. In the field of DRL, the DDPG algorithm is a better way to solve this problem, which can select the only deterministic action from the continuous action space in the continuous state space. Therefore, the structure of PORF-DDPG is shown in Figure 1.

In Figure 1, the dot-dashed area is used to provide the reward function with progressive optimization ability through human-machine collaboration. The DDPG model is used to learn autonomous driving behavior end-to-end. The progressively optimized reward function is used to provide dense rewards to adapt the DDPG model to different environments. At the same time, human supervision is used to ensure safe driving for the vehicle in the real environment and provide vehicle collision signals.

Considering the images of different environments or different weather and lighting in the same environment have a large gap in the texture, color, and element distribution, etc. The semantic segmentation images are used instead of the raw images as the model input to reduce the gap effectively. Figure 2 shows the comparison between the raw images and the semantic segmentation images of the two kinds of environments. Then, we use the DeepLab V3+ semantic segmentation algorithm [22] to provide semantic segmentation frames for model input in the real environment.

### 2.2. Reward Function with Progressive Optimization Capability

The reward setting of the reinforcement learning (RL) algorithm relates to the effectiveness and efficiency of model training. Improper reward setting may cause the model to fail in learning the correct results. Russell et al. [23] give an example: when the reward of a vacuum cleaner is set to “absorb dust”, vacuum cleaners will get rewards by “spraying dust” and then “absorbing dust”. The sparse reward may lead to inefficient model exploration and learning, slow iteration, and even difficulty in convergence. Such as in Go, from the beginning to play, the game can only be judged after the end of the game and the agent can get the reward at this time. In the middle of the game, the reward is difficult to evaluate. Therefore, it is necessary to ensure that the reward is reasonable and can accelerate the model training convergence. To prevent unmanned vehicles from doing too much useless exploration, a reward function with progressive optimization capability (PORF) is designed in this paper, which can provide a reward immediately after the vehicle completes an action. The reward function can be described as follows:(1)$$r=r^{\prime }+r_{DNN},$$
where $r^{\prime }$ represents the reward of the constraints on the vehicle driving states, it can be calculated in Equation (Equation 2):(2)$$r^{\prime }=k_{\alpha }max(|\alpha_{t}-\alpha_{t-1}|-\alpha_{T},0)+k_{C}|\alpha_{t}-\alpha_{C}|+k_{v}\left|v_{t}-v_{T}\right|+k_{Co}Co_{t},$$
where the first item $k_{\alpha }max(|\alpha_{t}-\alpha_{t-1}\left|-\alpha_{T},0\right)$ is used to limit the range of steering changes. When the difference between the steering commands given twice in succession is higher than the preset threshold $\alpha_{T}$, the vehicle will be given a penalty. This penalty can ensure that the steering policy will not make the vehicle oscillate greatly. The second item $k_{C}\left|\alpha_{t}-\alpha_{C}\right|$ is a restriction on the type of task performed by the vehicle. *C* is a high-level command given by humans, which indicates the type of tasks that the vehicle performs in the current environment. In this work, there are three kinds of vehicle tasks, including going straight, turning left, and turning right. For each task *C*, there is a corresponding reference steering direction $\alpha_{C}$. If the steering $\alpha_{t}$ executed currently is inconsistent with $\alpha_{C}$, there will be a penalty to the vehicle. This penalty mainly ensures that the vehicle can complete the turning task at the intersection. After obtaining the corresponding turning task at the intersection, it can turn according to the reference direction $\alpha_{C}$. The third item $k_{v}\left|v_{t}-v_{T}\right|$ provides the vehicle with an optimal driving speed $v_{T}$. The vehicle can be punished when the current speed $v_{t}$ is inconsistent with $v_{T}$ so that the vehicle speed can remain stable. The last item $k_{Co}Co_{t}$ provides punishment for the vehicle if it collides in a virtual environment or performs an unreasonable action in the real environment. This punishment provides a basis for vehicle steering and obstacle avoidance to ensure driving safety of the vehicle. $k_{\alpha }$, $k_{C}$, $k_{v}$ and $k_{Co}$ are the corresponding proportional coefficients of each reward item.

In Equation (Equation 1), $r_{DNN}$ is used to describe the comprehensive evaluation of vehicle driving behavior in the past period, which may be related to the vehicle-road relationship, pedestrian, obstacles, and other unforeseen factors. Considering that it is difficult to use a deterministic formula to describe these factors reasonably and accurately, a DNN reward model is built to regress the corresponding reward.

The DNN reward model structure designed by us is shown in Figure 3.

The DNN reward model training is divided into two stages: the pre-training stage and the progressive optimization stage.

*The pre-training stage*: The purpose of pre-training is to make the model have a preliminary ability to evaluate vehicle driving behavior based on the sequential images. In this stage, we need to build a training sample set $\left\{\langle \left(I_{i}^{1},\ldots ,I_{i}^{m}\right),Y_{i}\rangle |i=0,1,2,\ldots ,N\right\}$, where $\left(I_{i}^{1},\ldots ,I_{i}^{m}\right)$ represents the sequential images (the value of *m* is 5 in this paper), $Y_{i}$ represents the label of the sequential images which can be determined by experts based on various factors. Specifically, as shown in Figure 4, a feasible labeling method is that the label $Y_{i}$ can be determined by the vehicle-road relationship reflected in the image $I_{i}^{m}$, which can be described as follows:(3)$$Y_{i}=k_{1}\Delta d+k_{2}\Delta e,$$
where Δd indicates the lateral offset distance between the vehicle and the road’s centerline, and Δe indicates the angle between the vehicle’s orientation and the direction from the vehicle to 10 m ahead of the road’s central axis, $k_{1}$ and $k_{2}$ are proportional coefficients.

*The progressive optimization stage:* The purpose of this stage is to enable the DNN reward model to be continuously optimized via active human supervision and intervention. To reduce the pressure of human supervision and the difference in evaluation criterion between different people, human only needs to give a fuzzy evaluation $E\in \left\{high,normal,low\right\}$ for the pre-trained model output in the real environment. Then the evaluations from humans will be used to update the sample set automatically, which is used to train the DNN model to improve its evaluation accuracy progressively.

Algorithm 1 presents a human-in-the-loop progressive optimization method for the DNN reward model.
**Algorithm 1:** Progressive optimization algorithm of the DNN reward model under the human-machine collaboration.
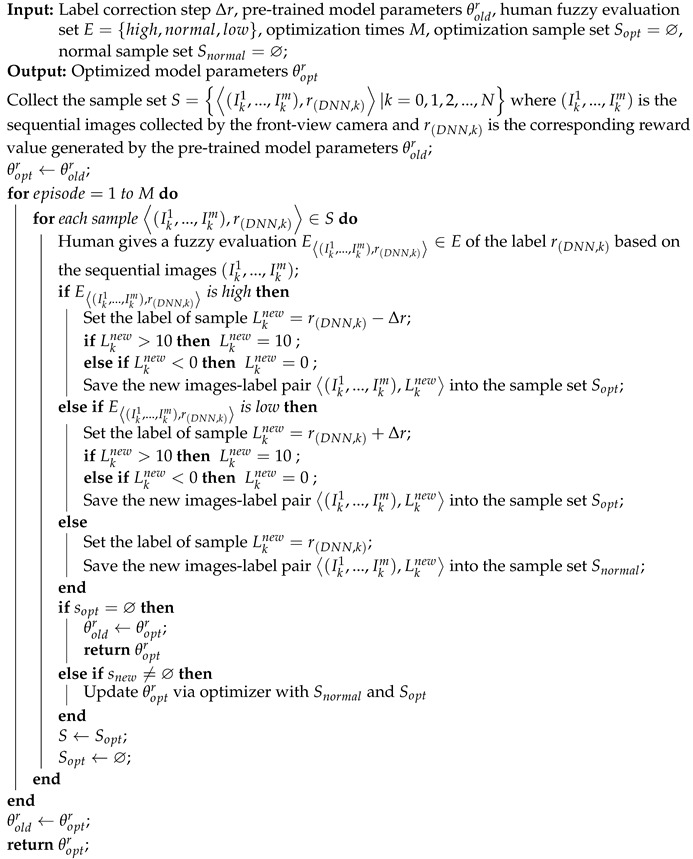


In Algorithm 1, line 2 describes the process of collecting sample set *S* in the real environment, lines 5 to 19 describe how to update the sample set *S* to obtain the optimization sample set $S_{opt}$ and the normal sample set $S_{normal}$ based on human fuzzy evaluation, and line 24 describes that the DNN reward model is optimized via feeding the updated sample set $S_{opt}$ and $S_{normal}$, then, line 21 and line 30 describe the optimized model parameters $\theta_{opt}^{r}$ are saved as the pre-trained model parameters $\theta_{old}^{r}$ which will be the initial model parameters in the next progressive optimization process.

### 2.3. Autonomous Driving Behavior Learning Model in PORF-DDPG

As mentioned in Section 2.1, an autonomous driving behavior learning model based on the DDPG algorithm is built as the end-to-end control model for unmanned vehicles, which can generate control policies directly from the vehicle state. The DDPG algorithm uses the actor–critic network structure, which is trained with memory samples to update the parameters of the model. Each memory sample is composed of four elements $<s_{t},a_{t},r_{t},s_{t+1}>$, where $s_{t}=\left\{I_{t},\alpha_{t},v_{t},C_{t}\right\}$ represents the vehicle state at the current moment, including a frame of semantic segmentation image $I_{t}$, steering $\alpha_{t}$, velocity $v_{t}$, and command $C_{t}$; $a_{t}$ represents the action performed by the vehicle in $s_{t}$; $r_{t}$ represents the reward obtained after performing the action $a_{t}$, and its value is calculated by PORF introduced in Section 2.2, and $s_{t+1}$ is the vehicle state of the next moment after completion of the action $a_{t}$.

In this paper, the actor–critic network structure designed by us is shown in Figure 5. Where the critic network is a Q network whose inputs are $s_{t}$ and $a_{t}$ and output is $Q(s_{t},a_{t})$. The Q network is responsible for estimating the expected value of the cumulative reward $Q(s_{t},a_{t})$ for performing the action $a_{t}$ in the state $s_{t}$. Through training, the critic network will satisfy the Bellman equation:(4)$$Q\left(s_{t},a_{t}\right)=E\left[r_{t}+\gamma Q\left(s_{t+1},\pi \left(s_{t+1}\right)\right)\right],$$
where γ is the discount factor of future rewards, the larger the γ, the more it values future rewards, and π(•) represents the policy that selects action *a* in state *s*.

The actor network is a policy network whose input is the state $s_{t}$ and output is the action $a_{t}$. It is an approximation of the optimal action policy $\pi^{*}\left(s\right)$ which can select the action that maximizes $Q(s_{t},a_{t})$ with the state $s_{t}$:(5)$$\pi^{*}\left(s_{t}\right)=argmax_{a_{t}}Q\left(s_{t},a_{t}\right)$$

In addition to using the actor–critic network structure, the DDPG model also uses two techniques. One is the experience replay. During the vehicle exploration in the environment, the samples collected have a strong successively correlation, which is not conducive to DDPG model training because it requires independent and identical distribution of the input data. Experience replay is used to make the input data of the DDPG model independent from each other. Specifically, experience replay is used to store the data that the vehicle interacts with the environment into a sample pool U(D) and then selects samples randomly to train the model.

The other technique is that DDPG creates two neural network copies for the policy network and the Q network, which are called online network and target network, respectively. Considering that in Equation (Equation 4), if we use the same critic network parameters to calculate $Q(s_{t},a_{t})$ and $Q(s_{t+1},\pi \left(s_{t+1}\right))$, there will be a correlation between them, which may cause training to shock. Therefore, the target Q network is introduced to calculate $Q(s_{t+1},\pi \left(s_{t+1}\right))$, where $\pi (s_{t+1})$ is calculated by the target policy network. Correspondingly, the online Q network is used to calculate $Q(s_{t},\pi \left(s_{t}\right))$, where $\pi (s_{t})$ is calculated by the online policy network.

During the training process of the DDPG model, after the online network parameters are updated, the target network parameters are soft updated through the running average algorithm as follows:(6)$$\theta^{Q^{\prime }}\leftarrow \sigma \theta^{Q}+\left(1-\sigma \right)\theta^{Q^{\prime }},$$
(7)$$\theta^{\pi^{\prime }}\leftarrow \sigma \theta^{\pi }+\left(1-\sigma \right)\theta^{\pi^{\prime }},$$
where $\theta^{Q^{\prime }}$ and $\theta^{Q}$ are the parameters of the target Q network and the online Q network respectively, $\theta^{\pi^{\prime }}$ and $\theta^{\pi }$ are the parameters of the target policy network and the online policy network respectively, and σ is the update rate of networks parameters.

After introducing the experience replay and the target networks, Equation (Equation 4) can be rewritten as:(8)$$Q_{\theta^{Q}}\left(s_{t},a_{\left(t,\theta^{\pi }\right)}\right)=E_{\left(s_{t},a_{t},r_{t},s_{t+1}\right)∼U\left(D\right)}\left[r_{t}+\gamma Q_{\theta^{Q^{\prime }}}\left(s_{t+1},\pi_{\theta^{\pi^{\prime }}}\left(s_{t+1}\right)\right)\right].$$

The Q network optimization goal is to minimize the error of the Bellman equation, which is called Temporal Difference (TD) error:(9)$$TD=\left[r_{t}+\gamma Q_{\theta^{Q^{\prime }}}\left(s_{t+1},\pi_{\theta^{\pi^{\prime }}}\left(s_{t+1}\right)\right)\right]-Q_{\theta^{Q}}\left(s_{t},a_{\left(t,\theta^{\pi }\right)}\right),$$
and the actor network optimization goal is to maximize the expected value of the cumulative reward $Q_{\theta^{Q}}\left(s_{t},a_{\left(t,\theta^{\pi }\right)}\right)$.

According to the above, Once the DDPG model training is completed, the model can determine the correct driving behavior of the vehicle based on the current input states. The driving behavior includes two DOF of steering control command and speed control command.

In addition, each DOF is normalized to the interval [−1, 1] in this paper. −1 means the steering wheel is turned to the far left or the brake pedal is at the bottom, 1 means the steering wheel is turned to the far right or the throttle is at the bottom, while 0 means not to turn the steering wheel or not to use the throttle and the brake.

### 2.4. Training Process of the Autonomous Driving Behavior Learning Model

As shown in Figure 6, the training process of the autonomous driving behavior learning model follows the principle of migrating from the virtual environment to the real environment. The entire model training process is divided into four steps. The first two steps are to pre-train the DNN reward model and the DDPG model in the virtual environment, and the last two steps are to progressively optimize the DNN reward model and the DDPG model through human-machine collaboration, which makes the vehicle learn driving behavior progressively in the real environment.

In step 1, we collect sequential images at equal time intervals in the virtual environment and label them by experts to generate the pre-train sample set $\left\{\langle \left(I_{i}^{1},\ldots ,I_{i}^{m}\right),Y_{i}\rangle |i=0,1,2,\ldots ,N\right\}$, which is used to train the DNN reward model. Then, sequential images are fed to the model to regress a reward value in the range [0, 10], which can be added with $r^{\prime }$ according to Equation (Equation 1) to provide dense reward $r_{t}$ for DDPG model training.

In step 2, in order to improve the efficiency of exploration, we pre-train the actor network through a supervised training method first so that the vehicle has a certain degree of autonomy. Thus, the vehicle will get the correct memory samples faster than the random exploration policy. The control policy given by the pre-trained actor network is used to control the vehicle and collect memory samples, which are used to train the actor network and the Q network to get a DDPG model that can make the vehicle run autonomously in the virtual environment.

In step 3, based on the Cityscapes dataset [24], the DeepLab V3+ semantic segmentation network [22] is trained first to provide semantic segmentation images in the real environment. Next, the sample set $S=\left\{\langle \left(I_{k}^{1},\ldots ,I_{k}^{m}\right),r_{\left(DNN,k\right)}\rangle |k=0,1,2,\ldots ,N\right\}$ is collected and then updated to the optimization sample set $s_{opt}$ and the normal sample set $s_{normal}$ via the human-machine collaboration method. Finally, the DNN reward model is trained based on $s_{opt}$ and $s_{normal}$ for progressive optimization, and the $r_{DNN}$ calculated by the optimized DNN reward model is added with $r^{\prime }$ obtained under the human active intervention to generate a more reasonable reward $r_{t}$ for the DDPG model training.

In the last step, the DDPG model is progressively optimized based on the more reasonable immediate reward $r_{t}$. Then, based on the optimized actor network in the DDPG model, a more reasonable policy is determined to drive the vehicle in the real environment.

We can repeat steps 3 and 4 in Figure 6 to deal with the migration of PORF-DDPG from one real environment to another. At this time, the optimized model parameters in the original environment will become the initial model parameters in the new environment, thus, the DNN reward model and the DDPG model will be iteratively optimized, respectively. In addition, the training of the two models is not strictly in order. When it is found that the DDPG model cannot learn the correct driving policy, more work should be carried out to optimize the DNN reward model further.

## 3. Results and Discussion

To verify the feasibility and effectiveness of PORF-DDPG in this paper, many experiments have been carried out. In this section, we introduce the experimental system and model training parameter setting first. Then, we carried out experimental verification and analysis of PORF-DDPG in the virtual environment and the real environment, respectively. Finally, we objectively discuss the experiment results of PORF-DDPG.

### 3.1. Experimental System and Model Training Parameters Setting

In this paper, the experiment was conducted in the CARLA open-source autonomous driving simulation platform and the real environment, respectively. The experiment platforms in the two environments are shown in Figure 7.

The training of the DNN reward model and the DDPG model was conducted on a computer configured with Intel(R) Core(TM) i5-9400 2.9 GHz CPU, 16 GB memory, and NVIDIA GTX1660Ti graphics card (6 GB video memory). The operating system was Ubuntu 16.04, and PyTorch 1.1 and Cuda 10.0 was used to build a neural network computing environment. The training parameter settings are shown in Table 1.

We set the parameters in Equation (Equation 2) based on a large amount of experimental experience, and the parameter settings are shown in Table 2.

In the virtual environment, the road is mainly two-lane, and the vehicle is expected to drive in the middle of the right lane. The real environment of this experiment mainly included two-lane and single-lane. when driving in the two-lane, the vehicle was expected to drive in the middle of the right lane like in the virtual environment. When driving on a single lane, we intentionally let the vehicle learn to drive on the right side of the lane instead of in the center, which reflected our personalized driving behavior. The expected driving trajectory is shown in Figure 8.

### 3.2. Experiments in the Virtual Environment

#### 3.2.1. Training and Testing of the DNN Reward Model

We pre-trained the DNN reward model in the virtual environment shown in Figure 9a. The sequential images in A→B section and D→E section were collected by the camera on the virtual vehicle and labeled according to Equation (Equation 3) in Section 2.2. Thus, a data set $S=\left\{\langle \left(I_{i}^{1},\ldots ,I_{i}^{m}\right),Y_{i}\rangle |k=0,1,2,\ldots ,1252\right\}$ was created, of which 1080 samples were used for the training of the DNN reward model, and the rest were used for model testing. We verified the trained DNN reward model in B→C section. The experiment results of the DNN reward model are shown in Figure 10.

Figure 10a shows the loss curve on the training set and testing set, the blue line represents the training loss curve while the orange line represents the testing loss curve. As the training loss decreases, the testing loss reduces to about 1 point progressively, indicating that the average test error is about 1 point after the model converges. Figure 10b shows the output of the DNN reward model (blue line) and the ground truth (orange line) while verifying the model in B→C section. The serial number represents the order of the input sample. In most cases, the output of the DNN reward model is consistent with the ground truth, and there is a difference of about 1 point in a few cases and almost no difference greater than 3 points. The experiment results show that the DNN reward model can give correct evaluations of driving behavior according to the sequential images during the vehicle running.

#### 3.2.2. Training and Testing of the DDPG Model

We pre-trained the DDPG model in the virtual environment shown in Figure 9b. Based on PORF designed in Section 2.2. the DDPG model was expected to learn how to drive correctly on A→B section and A→F→G→H→E→B section. First, we collected vehicle states $s_{t}=\left\{I_{t},\alpha_{t},v_{t},C_{t}\right\}$ on the two sections and labeled them by experts. Thus, a data set containing 6787 samples was created, of which 5455 samples were used for the actor network pre-training and the rest were used for the actor network testing. The pre-training and testing results of the actor network are shown in Figure 11a. The experiment results show that the training loss and testing loss eventually converge to a position neighbor to 0, indicating that the pre-trained actor network could control the vehicle well in the virtual environment.

Then, the pre-trained actor network was used to control the vehicle on the two sections to collect experience replay memory, the actor network output was combined with noise to ensure sufficient exploration of the vehicle states, and the reward $r_{t}=r_{t}^{\prime }+r_{\left(DNN,t\right)}$ was calculated using Equation (Equation 1), where $r_{t}^{\prime }$ was generated using Equation (Equation 2) and $r_{\left(DNN,t\right)}$ was given by the DNN reward model. Therefore, an experience replay memory pool containing 32,515 memory samples was created to train the DDPG model. The training results of the actor network are shown in Figure 11b. As mentioned in Section 2.3, the actor network’s optimization goal is to maximize the Q value and the critic network’s optimization goal is to minimize the TD error. In Figure 11b, as the TD error decreases, the Q value increases slowly, which indicates that the critic network can evaluate the accurate Q value and the actor network can also choose the driving behavior with a higher Q value.

Finally, the DDPG models from different training episodes were tested on A→F→G→H→E→B sections in Figure 9b, we recorded the average of the cumulative reward of each model during 10 trips and the test duration was 60 s. The test results are shown in Figure 11c. The experiment results show that while the corresponding TD error of each model decreases progressively, the average cumulative reward shows an upward trend, indicating that with the gradual enhancement of the fitting ability of the critic network, the actor network can choose actions with higher rewards according to the states and control the vehicle better.

### 3.3. Experiments in the Real Environment

To verify the progressive optimization ability of PORF-DDPG in the real environment, we have experimented many times in the campus environment with a small four-wheeled unmanned vehicle (as shown in Figure 7b). The experimental scenario is shown in Figure 12, where A→B section and B→C section are similar to the virtual environment in Section 3.2, which are two-way single-lane roads. However, Unlike the virtual environment in Section 3.2, C→D section and D→A section are single-lane roads and narrower, and C→D section varies in width.

#### 3.3.1. Progressive Optimization of the DNN Reward Model

To verify the feasibility and effectiveness of Algorithm 1 proposed in Section 2.2 which can progressively optimize the DNN reward model in the real environment, the sequential images of the four road sections were collected by the camera on the real vehicle which was remotely controlled by the human, and labeled according to the pre-trained DNN reward model. Thus, a data set $S=\left\{\langle \left(I_{k}^{1},\ldots ,I_{k}^{m}\right),r_{\left(DNN,k\right)}\rangle |k=0,1,2,\ldots ,19,493\right\}$ was created, of which 12,562 samples were used for the progressive optimization of the DNN reward model, and 6932 samples were used for testing. Among the test samples, half of the samples $S_{midtest}$ were from the middle of the lane and the other half $S_{edgetest}$ were from the edge of the lane. It should be noted that in the process of collecting training samples with the unmanned vehicle remotely controlled by the human, there were no special requirements for the driving trajectory of the unmanned vehicle, which meant that the unmanned vehicle could be controlled to drive along the middle of the road or deviate from the centerline. In the process of collecting testing samples, as shown in Figure 8, the human needed to control the vehicle to drive along the center or the edge of the four road sections as much as possible, so that the collected samples can be given higher or lower rewards if the DNN reward model was effectively optimized. In the process of the experiment, the optimization times M=5, and the label correction step Δr=5,4,3,2,1 in each optimization time. The testing results of the DNN reward model are shown in Figure 13.

Figure 13a shows the average evaluation comparison among the models with different episodes of human-machine collaboration on the testing set, with the increase of the episodes, the average evaluations increase progressively. Figure 13b shows the evaluations of the pre-trained DNN reward model in the virtual environment (blue line) and the optimized model with 5 episodes of human-machine collaboration (orange line) on $S_{midtest}$. The pre-trained model has a certain effect on the samples from A→B section and B→C section which are similar to the virtual environment, and the corresponding evaluations are in a higher range and close to the optimized model reward. However, on the samples from C→D section and D→A section that are different from the virtual environment, the pre-trained model has a bad effect and the corresponding evaluations are in a low reward range. In contrast, the optimized model’s evaluations are in a higher reward range overall. Figure 13c shows the output rewards comparison of the optimized DNN reward model on $S_{midtest}$ (orange line) and $S_{midtest}$ (blue line). The rewards given by the optimized reward model on these two types of samples are significantly different, indicating that the optimized DNN reward model can accurately determine the driving behavior of the vehicle based on the sequential images.

Further analysis found that most of the rewards of the optimized DNN reward model in Figure 13b,c are in a high range, but there are still a few lower rewards. In response to this phenomenon, we checked the corresponding road image samples and found that the main reason was that the samples were collected by the remotely controlled vehicle, it was impossible to strictly guarantee that the vehicle has been walking in the center of the lane. Therefore, the corresponding rewards of the samples deviating from the lane center were lower.

The experiment results show that the DNN reward model can be progressively optimized via Algorithm 1 proposed in this paper. In the real environment, when the DNN reward model has been progressively optimized, a progressively optimized reward function $r=r^{\prime }+r_{DNN}$ can be used to provide dense rewards for the DDPG model.

#### 3.3.2. Learning Personalized Driving Behavior in the Real Environment

With PORF is integrated into the DDPG framework, the DDPG model was trained on the four sections shown in Figure 12. We used the DDPG model trained in the virtual environment to control the vehicle with noise to collect the memory samples $<s_{t},a_{t},r_{t},s_{t+1}>$ in the real environment. Then, an experience replay memory pool U(D) was created, including samples collected in the real environment and part of samples with distinctive features in the virtual environment. The memory pool was used to train the DDPG model. We repeated the above process using the optimized DDPG model to collect new samples for iterative optimization according to Section 2.4.

Four rounds of sample collection and model training processes were conducted in the real environment. Finally, an experience replay memory sample pool U(D) containing 59,890 samples was formed.

The pre-trained DDPG model in the virtual environment and the four optimized models in the real environment were used to control the vehicle without noise on the four sections in Figure 12, and we recorded the corresponding path and cumulative rewards of each model. The results are shown in Figure 14 and Table 3.

For easy observation, we show three paths corresponding to the human (the red dotted line), the pre-trained DDPG model (yellow line), and the 4th optimized DDPG model (green line) respectively in each subfigure in Figure 14. Both models completed the autonomous driving task in A→B section and B→C section. However, in C→D section and D→A section, the pre-trained model did not complete the task and was stopped by the human when it was about to hit the shoulder of the road, while the optimization model completed the task well. It is shown that the 4th optimized DDPG model could complete the task better in the real environment.

In Table 3, we can further see that as the round of sampling and training increased, the cumulative reward of each model in each section shows an upward trend. At the same time, we should also see that cumulative rewards decrease occasionally in certain sections. The main reason is that neural network training is optimized for all samples, the performance is improved for most samples, but the performance decreases for a few samples. Therefore, it is impossible to improve the performance of all the sections continuously. The proportional coefficients $k_{\alpha }$, $k_{C}$, $k_{v}$, and $k_{Co}$ in Equation (Equation 2) are all negative (as shown in Table 2), leading to the result that the reward calculated by Equation (Equation 1) is also negative. Thus, cumulative rewards have negative values due to the setting of the reward value. The experiment results show that the DDPG model can learn the correct autonomous driving behavior progressively with PORF proposed in this paper in the real environment.

### 3.4. Discussion

In this paper, PORF-DDPG involves the training process and online inference process of multiple neural network models. The training time and online inference time of related models are shown in Table 4, where the training time and online inference time of the DNN reward model is 180 ms and 25 ms respectively; the training time of the DDPG reward model is 85 ms and the online inference time of the actor network and critic network in the DDPG model are both 3 ms; the online inference time and training time of the DeepLab V3+ semantic segmentation model selected in this paper are 406 ms and 20 ms respectively.

In practical applications, the end-to-end autonomous driving control process mainly includes the processing of data collected from sensors on the unmanned vehicle and the generation of the driving policy. In PORF-DDPG, data processing mainly involves semantic segmentation of images, and the driving policy is generated by the trained actor network in the DDPG model. Therefore, according to the experiment results in Table 4, when PORF-DDPG is used for end-to-end autonomous driving control, the entire control period is about 20 + 3 = 23 (ms), which can better meet the real-time requirements of the unmanned vehicle in real scenarios. In the data collection process, the time interval of the sequential images in this paper is 500 ms, and the online inference time of the DNN reward model is 25 ms. Therefore, the DNN reward model has enough time to calculate the reward of the previous 5 images before collecting a new image. In summary, PORF-DDPG can better meet the real-time requirements of autonomous driving and online data collection for the unmanned vehicle.

The method proposed in this paper mainly considers that humans are important resources in the development of autonomous driving systems. Unlike autonomous driving systems based on supervised learning [10,11,12,13,14], our method uses RL to obtain action policies, which is an optimization process that maximizes the expected cumulative reward. In the previous works that applied RL to real vehicles, [17,25] used various constraints to idealize the real environment and made it completely knowable like the virtual environment. Fully accurate GPS and fully accurate maps were used to obtain rewards, resulting in the lack of practical application capabilities. In [21], sparse rewards were used to train the RL model, resulting in weaker training signals and lower learning efficiency. Instead, the method proposed in this paper used the DNN to generate rewards only by image sensor information. Furthermore, we used human-machine collaboration to solve the rationality of rewards. Thus, our method could provide dense and immediate rewards to the RL model without strict experimental condition constraints.

It needs to be emphasized that the contribution of this paper is to realize a progressive optimization method of autonomous driving behavior with human-in-the-loop. We introduce the human-machine collaboration method to make the DNN reward model learn human evaluation criterion for the driving behavior of the vehicle, and thus make the DDPG model learn the personalized driving behavior of the human. Therefore, the use of PORF-DDPG can effectively reduce the cost of research and maintenance of unmanned vehicles. Specifically, when an unmanned vehicle equipped with PORF-DDPG is delivered to the user, by using the provided software with a friendly interface, users can progressively optimize the autonomous driving behavior of the vehicle by themselves, thus, the unmanned vehicle can adapt to the actual application scenarios better.

## 4. Conclusions and Future Work

This paper proposes a personalized autonomous driving behavior learning method PORF-DDPG with a progressively optimized reward function. We verify the feasibility and the effectiveness of PORF-DDPG through many experiments in the virtual environment and the real environment. The results show that by establishing a human-in-loop autonomous driving behavior learning system, PORF-DDPG is convenient for humans to give fuzzy evaluations for the accuracy of the DNN reward model based on their driving experience, and then realizes the progressive optimization of the DNN reward model through human-machine collaboration. At the same time, PORF-DDPG can also ensure the safe driving of the vehicle during the process of collecting memory samples, and enable the unmanned vehicle to achieve the ability to gradually optimize its autonomous driving behavior in different environments.

For the work of this paper, there are still some aspects that need further research. First, after the unmanned vehicle equipped with PORF-DDPG is transferred from one environment to another, the confidence of PORF-DDPG can be evaluated by the similarity between the old environment and the new environment, so that the unmanned vehicle can decide whether to execute the driving commands generated by PORF-DDPG. In addition, to improve the generalization ability of PORF-DDPG in different environments, this paper uses semantic segmentation images as the input of the model. However, we did not do more research on the semantic segmentation algorithm and just chose an existing semantic segmentation algorithm for experimental research. Obviously, the performance and efficiency of the semantic segmentation algorithm will affect the applicability of PORF-DDPG to a certain extent. Therefore, we will do further research on the semantic segmentation algorithm to optimize and perfect our work.

## Figures and Tables

**Figure 1 sensors-20-05626-f001:**
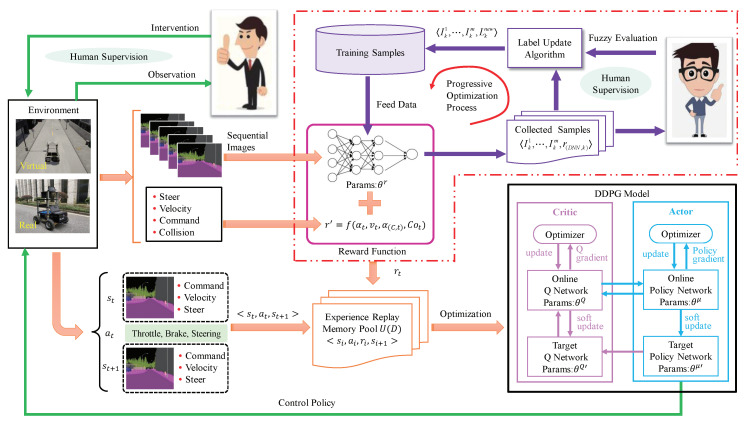
System structure of PORF-DDPG.

**Figure 2 sensors-20-05626-f002:**
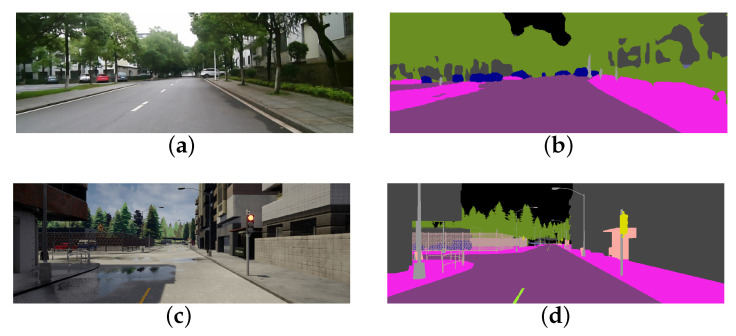
The comparison between the raw images and the semantic segmentation images. (**a**) is the raw image in the real environment, and (**b**) is the semantic segmentation image of (**a**). (**c**) is the raw image in the virtual environment, and (**d**) is the semantic segmentation image of (**c**). It can be clearly seen from the figures that the gap between the raw images of different environments is large, and the semantic segmentation image is an image obtained by coloring each type of element in the raw image uniformly, so the gap between the two is relatively small.

**Figure 3 sensors-20-05626-f003:**
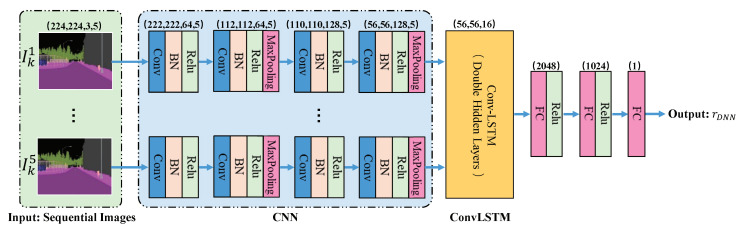
The DNN structure of the reward model. The inputs of the reward model are sequential images whose size is (224,224,3,5), and they are fed into the CNN to generate a feature map with the size of (56,56,128,5), then the feature map is fed into Conv-LSTM to extract the timing features among the sequential images which will be input to the fully connected layer to generate the output $r_{DNN}$. CNN uses the first six layers of the VGG16 (BN) network and loads the corresponding pre-training parameters.

**Figure 4 sensors-20-05626-f004:**
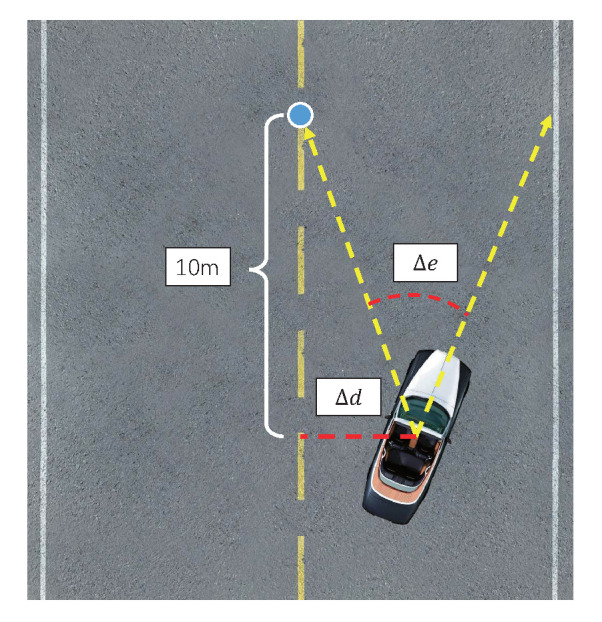
Schematic diagram of generating labels from the vehicle-road relationship.

**Figure 5 sensors-20-05626-f005:**
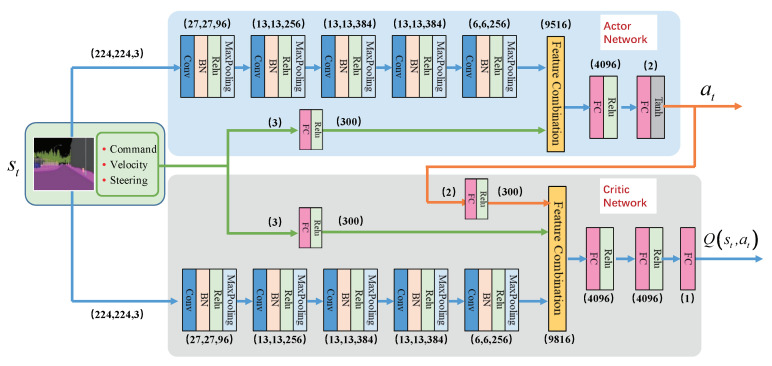
The actor–critic network structure in PORF-DDPG. For the input state $s_{t}$, the image whose size is (224,224,3) is fed into a 5-layer CNN to generate a feature map whose size is (6,6,256), and the 3-dimensional vector <command, speed, steering> is expanded to a 300-dimensional vector through a fully connected neural network. For the actor network, the expanded vector is combined with the feature map to generate the action $a_{t}=<steering\;control,speed\;control>$ through a fully connected neural network. The critic network has one more input action $a_{t}$ than the actor network. Similar to the actor network, the features extracted from the input state $s_{t}$ and the action $a_{t}$ are combined and fed into the fully connected neural network to obtain $Q(s_{t},a_{t})$.

**Figure 6 sensors-20-05626-f006:**
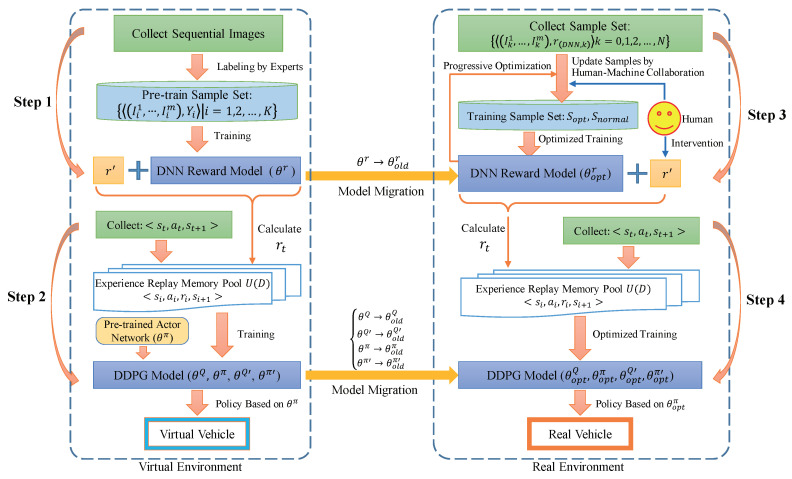
The training process of the autonomous driving behavior learning model.

**Figure 7 sensors-20-05626-f007:**
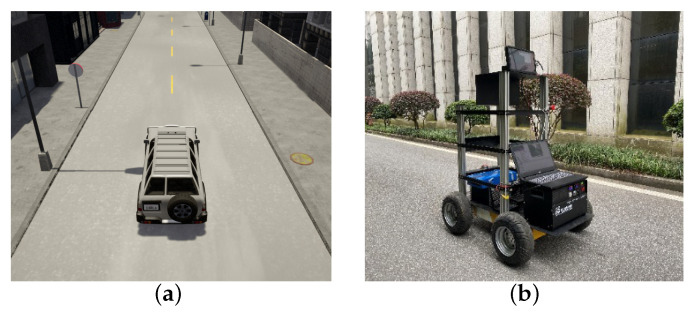
Experiment platform. (**a**) is a virtual vehicle used to collect data in the virtual environment built with CARLA. It was controlled by the DDPG model to learn autonomous driving behavior and equipped with a camera that could provide semantic segmentation images, collision sensors and pose sensors, etc. (**b**) is a small four-wheeled unmanned vehicle used as the experiment platform which could collect the front-view images by a monocular camera and record the vehicle trajectory by the encoder and IMU in the real environment.

**Figure 8 sensors-20-05626-f008:**
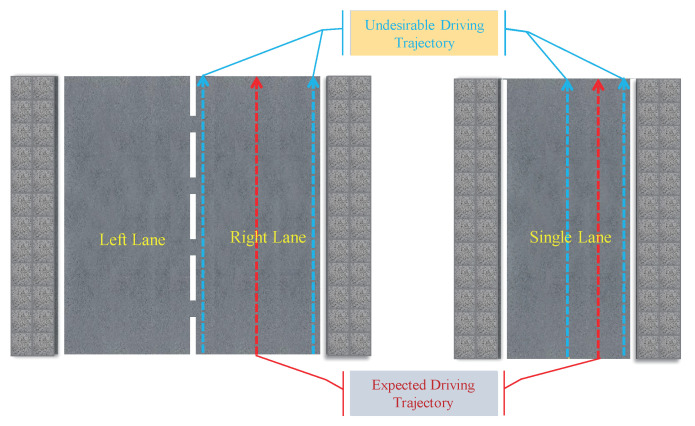
Schematic diagram of expected trajectories and undesirable driving trajectories. The red dotted arrow represents the trajectories we expect the vehicle to drive on this kind of road, and the blue dashed arrow represents the trajectories we do not expect the vehicle to drive on this road. At the same time, we will collect samples on these two types of trajectories for comparison experiments.

**Figure 9 sensors-20-05626-f009:**
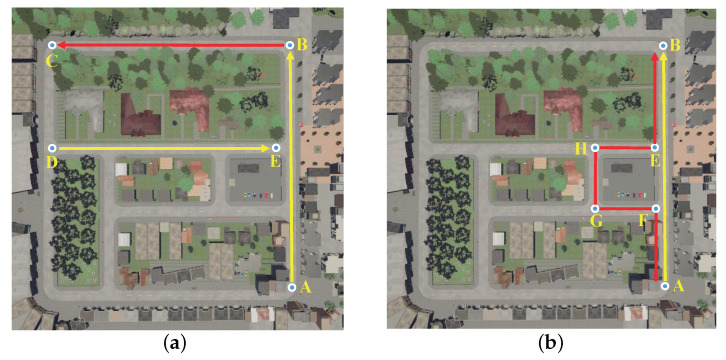
The virtual experiment environment. In subfigure (**a**), A→B section and D→E section were used for the training and testing of the DNN reward model, and B→C section was used to verify the effect of the trained DNN reward model. In subfigure (**b**), A→B section and A→F→G→H→E→B section were used to train and test the DDPG model.

**Figure 10 sensors-20-05626-f010:**
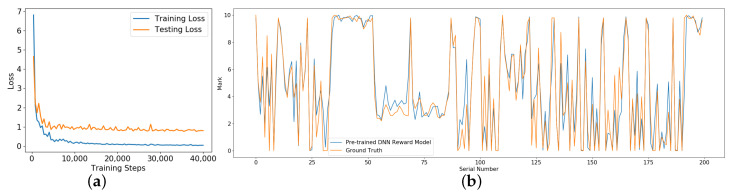
The performance of the DNN reward model in the virtual environment. (**a**) shows the loss curve on the training set and testing set; (**b**) shows the output of the DNN reward model (blue line) and the ground truth (orange line) while verifying the model in B→C section.

**Figure 11 sensors-20-05626-f011:**
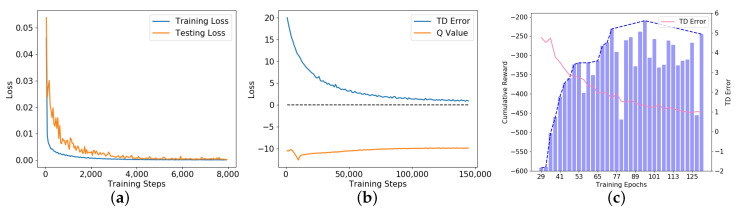
The performance of the DDPG model in the virtual environment. (**a**) shows the loss curve of the actor network pre-training, the blue line represents the training loss curve while the orange line represents the testing loss curve. (**b**) shows the loss curve of the DDPG model training, the blue line represents the TD error while the orange line represents the Q value. (**c**) shows the relationship between the TD error (red line) and the average cumulative reward (blue bar).

**Figure 12 sensors-20-05626-f012:**
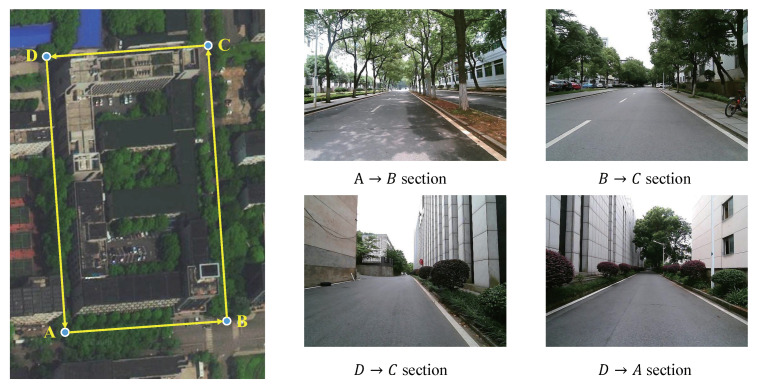
The real experimental environment.

**Figure 13 sensors-20-05626-f013:**
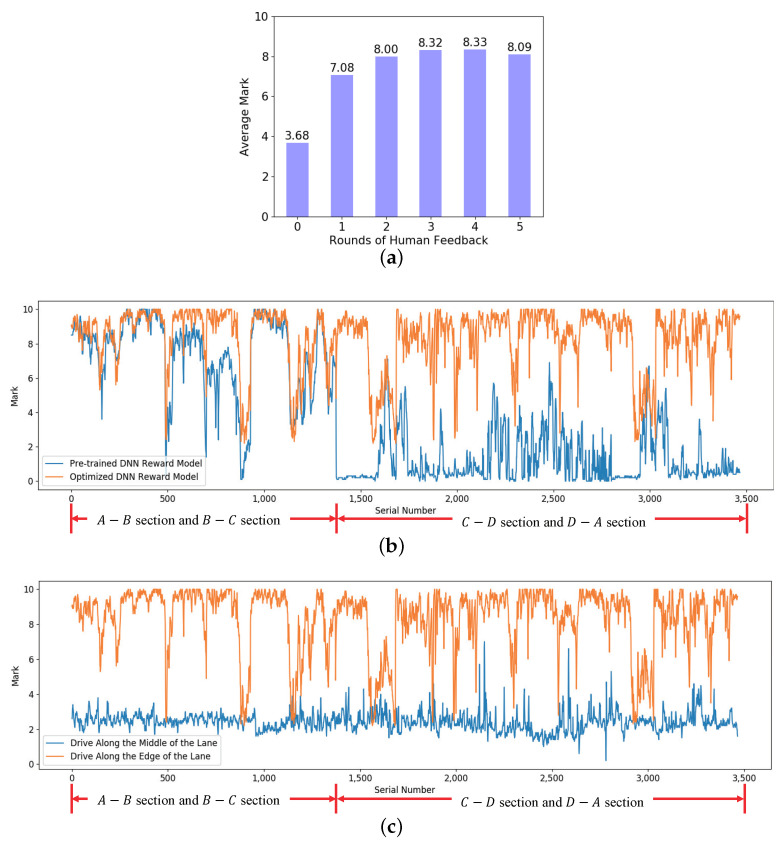
The performance of the DNN reward model in the real environment.

**Figure 14 sensors-20-05626-f014:**
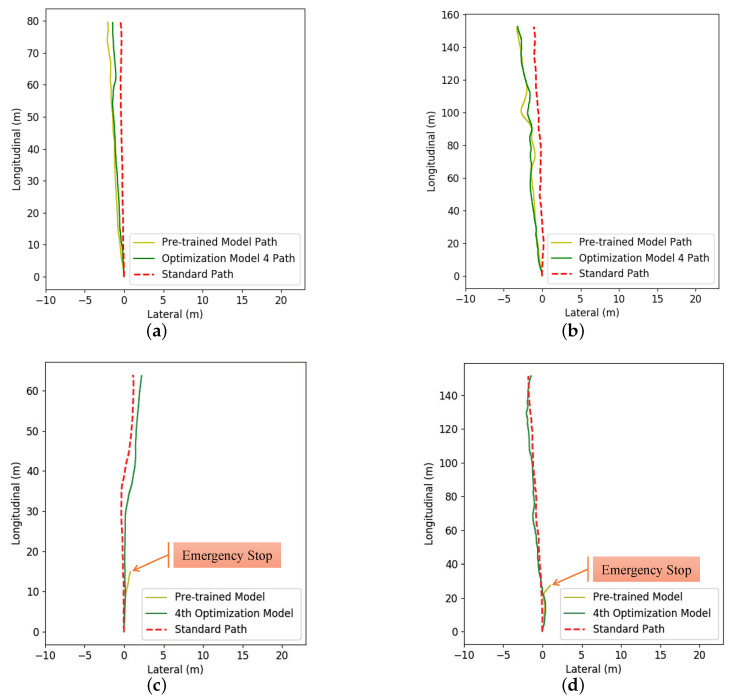
(**a**,**b**) are the paths of vehicle driving on A→B section and B→C section, (**c**,**d**) are the paths of the vehicle driving on C→D section and D→A section.

**Table 1 sensors-20-05626-t001:** Training parameters in the experiment.

Model	Loss Function	Optimizer	Learning Rate	Weight Decay
DNN Reward Model	MSE	Adam	1 × 10${}^{-4}$	1 × 10${}^{-5}$
Actor Network Pre-training	MSE	Adam	1 × 10${}^{-6}$	1 × 10${}^{-5}$
DDPG Model	MSE	Adam	1 × 10${}^{-5}$	1 × 10${}^{-5}$

**Table 2 sensors-20-05626-t002:** Parameter settings in Equation (Equation 2).

Parameter	Value
$\alpha_{T}$	0.2
$\alpha_{C}$	Going Straight: 0
	Turning Left: 0.35
	Turning Right: 0.5
$v_{T}$	15 km/h
$k_{\alpha }$	−10
$k_{C}$	−10
$k_{v}$	−10
$k_{Co}$	−30

**Table 3 sensors-20-05626-t003:** Cumulative rewards on each section.

Experiment Model	A→B Section	B→C Section	C→D Section	D→A Section
Pre-Trained Model	−490	−775	Unfinished	Unfinished
1st Optimization Model	−107	−711	Unfinished	Unfinished
2nd Optimization Model	−149	−697	−540	Unfinished
3rd Optimization Model	−137	−686	−725	−661
4th Optimization Model	−126	−671	−558	−682

**Table 4 sensors-20-05626-t004:** Model training time and inference time of PORF-DDPG.

Model	Training Time	Inference Time
DNN Reward Model	180 ms (Batch Size = 2)	25 ms
DDPG Model	85 ms (Batch Size = 10)	Actor Network: 3 ms Critic Network: 3 ms
The DeepLab V3+ Model	406 ms (Batch Size = 2)	20 ms

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
