# Peer review of "PORF-DDPG: Learning Personalized Autonomous Driving Behavior with Progressively Optimized Reward Function"

_sensors, 2020, doi:10.3390/s20195626_

Round 1

Reviewer 1 Report

This paper proposes a human-in-the-loop DRL algorithm to learn autonomous driving behavior in a progressive way. It introduces a deep nerual network to learn an optimized reward function based on human supervision and uses the DDPG framework to learn autonomous driving behavior. A variety of experiments have been conducted both in virtual and real-world environments to verify the effectiveness of the proposed framework. Generally, the topic is interesting and the paper is well organized. Some minor comments are given below.
1. The full name of "DDAC" should be given.
2. The figures need to be enlarged.
3. The lane in Fig.2 disappears after semantic segmentation. Please explain if there raises any problem.
4. Please explain the symbols in Eq.2 and how the thresholds are set.
5. Please explain how to obtain the offset distance and angle in Eq. 3.
6. Please explain line 6-15 in algorithm 1.
7. What is the difference between command Ct and action at?
8. How is the ground truth acquired in Fig. 10?
9. Do you mean you use raw images in step 1 while semantic segmentation images in step 3?
10. Correct typo "in certain sections" (line 392).

Author Response

Attached please kindly find the revised manuscript of our paper sensors-943353 to Sensors, entitled “PORF-DDPG: Learning Personalized Autonomous Driving Behavior with Progressively Optimized Reward Function”, by Jie Chen, Tao Wu, Meiping Shi and Wei Jiang.

Before starting our discussion on the changes incurred to the aforementioned paper, we would like to thank the Associate Editor and the anonymous Reviewers for their effort to evaluate and improve this manuscript. We greatly appreciate the positive comments from the anonymous reviewers and the associate editor. These comments have helped significantly improve the quality of our manuscript.

We hope that all of the comments and criticisms have been adequately responded to.

In the revised version, we have carefully revised the manuscript to meet the format requirements for publication. The revised texts in the new version of the manuscript are marked in blue.

Please feel free to let us know should you have any questions. Thank you!

Yours sincerely,

Corresponding author: Mrs. MeiPing Shi

E-mail: shimeiping@nudt.edu.cn

Reviewer 2 Report

The authors in their work consider a very interesting approach to the use of deep neural networks for autonomous driving problems.

The paper provides an extensive introduction containing relevant literature sources.

The introduction also formulates the contribution and novelty of the proposed approach.

To improve understanding of the work, I would like the authors to take into account the following considerations:

1) Is there noise filtering for camera images?

2) what is the basic way to segment images used by authors?

3) three tasks of the car movement are indicated (left, straight and right), is it possible to use the approach for moving back?

4) what will be the actions of the car if it is not possible to find the center line on the road (possibly due to weather conditions)?

5) what will be the actions of the car if a static (or dynamic) obstacle suddenly appears on the road?

6) what does gamma mean in equation (4)?

7) what is the size of the deep neural network (how many layers, neurons, outputs)?

8) in Figure 10 (b) what does the serial number mean (this is the number of experiments)?

9) in table 2 why cumulative rewards have negative values?

10) the experiments were carried out in good visibility conditions, were the experiments performed in poor lighting?

Despite this list of questions, I believe that the authors have done interesting and practically demanded work.

Author Response

(The authors gave the same response as above.)

Reviewer 3 Report

The authors of the article present a very interesting study and a partial solution to the problem encountered in autonomous vehicles. It is now an important scientific trend to support everyday life and strive for the greatest possible support for the driver during the journey.
The authors very precisely show the state of knowledge in this field, which proves an in-depth analysis of the current needs of this field of science.
The paper presents a proprietary algorithmic solution. The algorithm is precisely and clearly described. The mathematical apparatus used is precisely marked and commented out.
It should be emphasized that the model was verified both in virtual and real form with a test vehicle.
Of course, the obtained results are related to a certain "ideal state", so it is a long way from a universal implementation, but in my opinion they are a very interesting and important contribution to this subject. The results are promising and the authors rightly set the scope for further work.

Author Response

Attached please kindly find the revised manuscript of our paper sensors-943353 to Sensors, entitled “PORF-DDPG: Learning Personalized Autonomous Driving Behavior with Progressively Optimized Reward Function”, by Jie Chen, Tao Wu, Meiping Shi and Wei Jiang.

Before starting our discussion on the changes incurred to the aforementioned paper, we would like to thank the Associate Editor and the anonymous Reviewers for their effort to evaluate and improve this manuscript. We greatly appreciate the positive comments from the anonymous reviewers and the associate editor. These comments have helped significantly improve the quality of our manuscript.

We hope that all of the comments and criticisms have been adequately responded to.

In the revised version, we have carefully revised the manuscript to meet the format requirements for publication. The revised texts in the new version of the manuscript are marked in blue.

Please feel free to let us know should you have any questions. Thank you!

Yours sincerely,

Corresponding author: Mrs. MeiPing Shi

E-mail: shimeiping@nudt.edu.cn

This manuscript is a resubmission of an earlier submission. The following is a list of the peer review reports and author responses from that submission.